# The Complexities of Organ Crosstalk in Phosphate Homeostasis: Time to Put Phosphate Sensing Back in the Limelight

**DOI:** 10.3390/ijms22115701

**Published:** 2021-05-27

**Authors:** Lucile Figueres, Sarah Beck-Cormier, Laurent Beck, Joanne Marks

**Affiliations:** 1Department of Neuroscience, Physiology and Pharmacology, Royal Free Campus, University College London, London NW3 2PF, UK; joanne.marks@ucl.ac.uk; 2CHU de Nantes, Université de Nantes, F-44042 Nantes, France; 3Inserm, UMR 1229, RMeS Regenerative Medicine and Skeleton, Université de Nantes, ONIRIS, F-44042 Nantes, France; sarah.beck@univ-nantes.fr (S.B.-C.); laurent.beck@inserm.fr (L.B.)

**Keywords:** phosphate transporter, Slc34, Slc20, phosphate homeostasis

## Abstract

Phosphate homeostasis is essential for health and is achieved via interaction between the bone, kidney, small intestine, and parathyroid glands and via intricate processes involving phosphate transporters, phosphate sensors, and circulating hormones. Numerous genetic and acquired disorders are associated with disruption in these processes and can lead to significant morbidity and mortality. The role of the kidney in phosphate homeostasis is well known, although it is recognized that the cellular mechanisms in murine models and humans are different. Intestinal phosphate transport also appears to differ in humans and rodents, with recent studies demonstrating a dominant role for the paracellular pathway. The existence of phosphate sensing has been acknowledged for decades; however, the underlying molecular mechanisms are poorly understood. At least three phosphate sensors have emerged. PiT2 and FGFR1c both act as phosphate sensors controlling Fibroblast Growth Factor 23 secretion in bone, whereas the calcium-sensing receptor controls parathyroid hormone secretion in response to extracellular phosphate. All three of the proposed sensors are expressed in the kidney and intestine but their exact function in these organs is unknown. Understanding organ interactions and the mechanisms involved in phosphate sensing requires significant research to develop novel approaches for the treatment of phosphate homeostasis disorders.

## 1. Introduction

Inorganic phosphate is essential for mammalian life and plays a key role in nucleic acid synthesis, adenosine triphosphate formation, and bone mineralization. Phosphate is mainly stored as hydroxyapatite (Ca_10_(PO_4_)_6_(OH)_2_) in bone and teeth, with relatively small amounts found in extracellular fluid (2.5–4.5 mg/dL) [1]. Four organs are involved in the maintenance of extracellular phosphate concentration: the bone, the kidney, the small intestine, and the parathyroid glands. They interact with each other and respond to changes in extracellular phosphate concentrations via intricate processes involving phosphate transporters, phosphate sensors, and three circulating hormones: parathyroid hormone (PTH), Fibroblast Growth Factor 23 (FGF23), and calcitriol (1.25(OH)_2_ vitamin D_3_). 

The use of numerous animal models has helped to unravel the complex process involved in the regulation of phosphate homeostasis; however, it is becoming apparent that insights gained from rodent models do not always extrapolate to the processes that occur in humans. In this review, we outline the mechanisms of phosphate homeostasis and how disruption in these processes can lead to severe morbidity in humans. We go on to discuss the current knowledge regarding crosstalk between the kidney, the parathyroid, the bone and the small intestine for maintaining phosphate balance, specifically focusing on the role of phosphate transporters and phosphate sensors in this process. Using information gained from rodent models and human diseases we highlight the similarities and differences observed regarding the complexities of phosphate balance in different species.

## 2. Phosphate Homeostasis in Health and Disease

Phosphate sensing can be considered the first step in the maintenance of phosphate homeostasis, resulting in local changes in phosphate transporter expression and function and in endocrine responses that coordinate subsequent interorgan crosstalk. Figure 1 outlines the complexities of this process, demonstrating the phosphate transporters and phosphate sensors present in each organ. It highlights the site of production and the site of action of the endocrine hormones involved in maintaining phosphate balance and the feedback loops that exist to provide constant crosstalk between the four organs. Interestingly, recent studies have identified the circadian rhythm, iron deficiency, anemia, metabolic acidosis, and inflammation as important novel regulators of FGF23, highlighting the fact that this already intricate process has additional layers of complexity (see Mace et al. for a recent comprehensive review [2]).

Derangement in any one of the steps involved in maintaining phosphate homeostasis can manifest as a significant clinical disorder. The most common is chronic kidney disease (CKD), which was reported by the Global Burden of Disease study in 2017 to have a global prevalence of 9.1% [4]. Morbidity is higher in patients with CKD who have hyperphosphatemia, particularly those on dialysis [5]. The elevated serum phosphate load stimulates vascular smooth muscle cells to undergo phenotypic changes that predispose them to calcification, ultimately leading to poor cardiovascular prognosis [6] and mortality [7] in these patients. In early CKD, the physiological adaptation of elevated FGF23 levels to maintain serum phosphate has also been associated with left ventricular hypertrophy, cardiovascular events, and all-cause mortality (for recent review see [8]). In addition to the vascular phenotype, CKD can also be associated with changes in bone morphology, bone density, and remodeling activity. Collectively, the systemic disorder of mineral and bone metabolism due to CKD has been defined by the Kidney Disease-Improving Global Outcomes (KDIGO) Foundation to include individual or in combination abnormalities in calcium, phosphorus, PTH, or vitamin D metabolism; abnormalities in bone turnover or mineralization; and vascular or other soft tissue calcification [9]. 

Maintaining phosphate balance may also be a challenge in healthy individuals ingesting the type of highly processed diet that is becoming prevalent in industrialized countries. Ingestion of these food types, where significant amounts of highly bioavailable phosphate-based preservatives are added to improve taste and increase shelf life, has led to an ongoing debate as to whether these preservatives impact human health outcomes [10,11]. While evidence for phosphate toxicity in humans might not be conclusive, a high dietary phosphate burden has been reported to induce significant changes in vascular calcification [8] and bone health in rodent models [12]. Recent rodent studies have also provided unexpected insights into the short- and long-term hormonal adaptions that occur in response to dietary phosphate load [12], which may change our understanding and perception of human disease outcomes. 

In addition to the prevalent condition of CKD, there are numerous rare genetic disorders that are associated with disturbances in phosphate homeostasis. These include, but are not limited to, diseases caused by mutations in the proteins responsible for renal phosphate reabsorption, and those associated with the regulation of FGF23 synthesis (reviewed in detail in [13]). Of the latter disorders, X-linked hypophosphatemic rickets (XLH) is the most common and is caused by mutations in the PHEX gene leading to elevated FGF23 secretion and subsequent hypophosphatemia. Collectively, this group of congenital conditions causes significant bone mineralization defects, manifesting as rickets and osteomalacia in children and osteomalacia in adults [13]. 

Therefore, because of the numerous diseases related to altered phosphate homeostasis and, in particular, the increasing prevalence and burden of chronic hyperphosphatemia due to CKD, in vitro and in vivo models have been developed to help unravel the complex mechanisms involved in maintaining phosphate balance. These models have provided important information regarding the role of phosphate transporters and sensors in this process and such knowledge should allow the development of targeted therapeutics to treat these diseases. 

## 3. Phosphate Transporters: Knowledge from Animal Models and Human Physiology

Phosphate balance in extracellular fluids is maintained by the regulation of renal phosphate reabsorption in proximal tubular cells, by adjustment of enterocyte phosphate absorption, and by the turnover of the bone mineral component (Figure 1). Two families of sodium phosphate co-transporters, SLC34 and SLC20, are present in the different cell types, and their expression pattern and function are critical for efficient phosphate balance. 

### 3.1. SlC34 Sodium Phosphate Co-Transporters 

NPT2a (SLC34A1) and NPT2c (SLC34A3) are expressed in the brush border membrane of renal proximal tubular cells [14]. The phosphaturic hormones, PTH and FGF23, act through their respective receptors, also expressed in this nephron segment, to decrease phosphate reabsorption. Numerous studies using cultured cells [15,16,17] and rodent models [15,18] have shown that this involves internalization of both isoforms of the SLC34 family. Interestingly, at least in response to PTH, there is acute internalization of NPT2a [17] and delayed internalization of NPT2c [15]. SlC34A1 inactivation in mice causes significant phosphaturia and hypophosphatemia, whereas, SlC34A3 inactivation is not associated with hypophosphatemia in mice [19], suggesting a predominant role of NPT2A in phosphate reabsorption in this model (Table 1). In contrast, inactivating mutations in SLC34A1 [20] or SLC34A3 [21] in humans are both associated with hypophosphatemia and nephrolithiasis (Table 1). Biallelic inactivating mutations in SLC34A1 may also induce infantile hypercalcemia [22] or Fanconi syndrome [23]. The recent study of a family who displayed heterozygous inactivating mutations of SLC34A1 and SLC34A3 with severe hypophosphatemia and rickets supports the synergic and non-redundant action of SLC34A1 and SLC34A3 on renal phosphate transport [24]. These findings highlight the need to consider animal model data with caution regarding the molecular mechanisms of renal phosphate handling in humans.

The difference between murine and human inactivation of SlC34 sodium phosphate co-transporters is also supported by the bone phenotype displayed by the two species (Table 1). SLC34A1 inactivation has been associated with possible bone defects in mice [19]. Some SLC34A1 mutations in humans have also been associated with osteoporosis [20]; however, it remains to be determined if this bone involvement is due to the chronic hypophosphatemia caused by the renal phosphate reabsorption impairment or if it is due to the specific inactivation of SLC34A1 expressed in bone [39]. In osteoblast-like cells, NPT2A is upregulated by an increase in extracellular phosphate [40], suggesting a role for this transporter in bone mineralization; however, the function of this protein in bone in vivo remains elusive [39]. To our knowledge, there is no significant evidence of NPT2C expression in bone [39] and SLC34A3 inactivation does not induce a bone phenotype in mice [31]. However, some mutations in humans are associated with severe osteoporosis or rickets (hereditary hypophosphatemic rickets with hypercalciuria) [21,29]. Much work is needed to elucidate whether there is a role of SLC34 sodium phosphate co-transporters in bone.

NPT2B (SLC34A2) is expressed in the enterocyte brush border membrane, where it is upregulated by calcitriol [41], and in alveolar type II cells in the lungs. The mRNA expression of NPT2B in human lung is 30-fold higher than in the small intestine [42,43]. NPT2B has also recently been localized in the kidney medullary thick ascending limb in humans, although levels are ~100-fold lower than in lung [42]. Whereas it has been shown that renal NPT2B transcripts are upregulated in a mouse model of chronic kidney disease (CKD) [43], its overall role in normal renal phosphate handling remains unknown. Diseases that have the potential to impact NPT2B function would be expected to cause an imbalance in phosphate homeostasis if this protein plays a critical role in intestinal phosphate absorption. However, in SLC34A2-intestinal-specific knockout mice, phosphatemia remains normal [27,28], and human inactivating mutations in SLC34A2 cause pulmonary alveolar microlithiasis, with an exclusive lung phenotype without hypophosphatemia [25] and without evidence of intestinal phosphate absorption alterations (Table 1). Similarly, hypercalcitriolemia disorders (i.e., granulomatosis or inactivating mutations in CYP24A1, the gene encoding 24-hydroxylase), which would be expected to enhance NPT2B function, are not classically associated with hyperphosphatemia [44]. Based on these findings and other supporting evidence, such as the inefficiency of NPT2B inhibitors to correct hyperphosphatemia in CKD [45], it is now becoming accepted that the paracellular pathway is likely to be the dominant route for intestinal phosphate absorption [46,47]. It is proposed that tight junctions, and in particular the claudin family of proteins, may determine phosphate permeability [48]. Although there is emerging evidence that changes in claudin 3 protein levels in response to the vitamin D receptor agonist lithocholic acid [48], and following dietary iron deficiency [49], impacts intestinal phosphate absorption in rodents, there is no data to confirm whether claudin expression impacts phosphate absorption in the human small intestine. As the main sites for intestinal phosphate significantly differ between species (duodenum and jejunum in humans and rats [50,51] and ileum in mice [51]), results from rodent models studying the processes of intestinal phosphate handling may not fully translate to humans.

### 3.2. Slc20 Sodium Phosphate Co-Transporters 

PiT1 (SLC20A1) and PiT2 (*SLC20A2*) are high-affinity sodium-dependent phosphate transporters [52,53] that were originally identified as retrovirus receptors [54,55,56]. Unlike sodium phosphate co-transporters from the SLC34 family, PiT proteins are widely expressed, including in key organs involved in phosphate homeostasis regulation [57]. In bone, PiT1 and PiT2 are expressed in chondrocytes and osteoblasts [34,58] but the actual contribution of these proteins to bone mineralization is still a matter of debate [59]. Both PiT1 and PiT2 have been detected in the kidney, but localization at the apical membrane of the proximal tubule has only been confirmed for PiT2 [14]. However, available experimental data suggest a negligible contribution of PiT2 to transcellular phosphate reabsorption, consistent with the predominant role of NPT2A AND NPT2C in this process [14]. Normophosphatemia in humans with SLC20A2-inactivating mutations [60] and SLC20A2 knockout mice [34] further support this hypothesis (Table 1). In addition to the bone and kidneys, PiT1 and PiT2 are also expressed in the intestine, specifically at the apical membrane of enterocytes [61,62]. However, the relative contribution of PiT1 and PiT2 to intestinal phosphate absorption is still a matter for deliberation. The expression of PiT1 is segment-specific and comparable to NPT2B, with strong expression in the duodenum and jejunum [62,63], whereas, PiT2 is expressed all along the intestine at low levels [64]. Historically, a role for the SLC20 transporters in transcellular sodium-dependent phosphate absorption by the intestine has generally been dismissed, as SLC34A2 deletion reduces transcellular phosphate absorption by ~90% [27]. This hypothesis is also consistent with the normal phosphate balance observed in intestinal-specific SLC20A2 knockout mice, without compensatory upregulation of NPT2B or PiT1 [65]. A recent study however, suggests that SLC20 sodium phosphate co-transporters may well participate in intestinal phosphate absorption, as the use of the pan inhibitor, EOS789, in a rat model of CKD, had a stronger effect on reducing hyperphosphatemia than the use of a NPT2B-specific inhibitor alone [66]. In addition, the same group has also reported that whereas the contribution of NPT2B to transcellular absorption is dominant under normal conditions, SLC20-associated transcellular phosphate transport is likely to play a key role in intestinal phosphate absorption in rats with CKD [64]. Based on the relative expression levels of SLC20 proteins, the authors suggest that PiT1 rather than PiT2 is responsible for the low affinity phosphate absorption observed in rats with CKD [64]. In contrast, the widespread and low level expression of PiT2 in epithelial and non-epithelial cells throughout the intestine [64], together with its questionable role in transcellular phosphate absorption, indicates that this protein may potentially function as an intestinal phosphate sensor [59].

## 4. Phosphate Sensing: The Knowns and Unknowns

Sensing the variations in extracellular phosphate concentrations and informing the cell of such changes is defined by the term “phosphate sensing.” Although the existence of such a mechanism has been acknowledged for decades, the underlying physiological, endocrine, cellular, and molecular mechanisms are poorly understood and are only beginning to be slowly deciphered. Phosphate sensing may involve a single cell, a whole organ, or a combination of different organs and factors. At the whole-body level, phosphate sensing is the first step of phosphate homeostasis regulation. When extracellular phosphate varies, key phosphate sensing cells may induce subsequent local or endocrine responses. For example, the idea that the intestinal epithelium can sense phosphate and signal to the kidney to maintain phosphate balance has been proposed. In 2007, Berndt et al. suggested that a gut-derived factor is released in response to ingestion of dietary phosphate and that this factor rapidly modulates renal phosphate reabsorption to prevent large post-prandial fluctuations in serum phosphate [67]. Subsequent studies have been unable to confirm the existence of this “intestinal phosphatonin” [68,69] and have instead found that acute renal adaptation to an oral phosphate load requires a parathyroid–kidney and a bone–kidney axis. FGF23, synthetized by osteocytes, and PTH, synthesized by the parathyroid glands, are secreted in response to the increase in extracellular phosphate [70] and act to decrease renal phosphate reabsorption and restore phosphate homeostasis [15,16,17]. Calcitriol, a third circulating factor produced by 1alpha hydroxylation of 25OH vitamin D in proximal tubular cells, is also involved in the regulation of extracellular phosphate. PTH increases calcitriol production, which in turn increases intestinal phosphate absorption (through NPT2B) [41], and thus may appear to be counterproductive to the maintenance of phosphate balance. This effect, however, is counterbalanced by the feedback loop of calcitriol on PTH synthesis, its positive effect on bone formation, and its stimulation of FGF23 synthesis [71]. These loops of regulation maintained by these three circulating factors involve at least three different phosphate sensor candidates: the sodium phosphate co-transporter PiT2, the calcium sensing receptor (CaSR), and the FGF receptor 1 (FGFR1c).

The role of PiT1 as a putative phosphate sensor was suggested a decade ago through its ability to mediate extracellular signal-regulated kinase (ERK) phosphorylation upon an increase in extracellular phosphate levels in vitro [72,73]. Later, it was found that the active phosphate sensor consisted of low-abundance PiT1-PiT2 heterodimers, the activity of which was dependent upon phosphate binding but not transport [74]. Despite this new knowledge, the physiological importance of such a sensor still remains to be illustrated, although PiT2 disruption in vivo fully blunts the secretion of FGF23 in response to variations in dietary phosphate loads, without changes in PTH or vitamin D levels [70]. Moreover, the same results were obtained using organ cultures of long bones from SLC20A2 knockout mice, further illustrating the possible role of PiT2 as a phosphate sensor in bone, acting to control phosphate-dependent FGF23 secretion. Importantly, this role of PiT2 as a phosphate sensor in vivo could only be illustrated upon phosphate challenge, a finding that is consistent with the normal FGF23 serum phosphate levels of SLC20A2 knockout mice under normal dietary phosphate conditions [34].

Interestingly, Pastor-Arroyo et al. [65] demonstrated that under phosphate restriction, FGF23 secretion was still able to adapt in intestinal-specific SLC20A2 knockout mice, suggesting that the role of PiT2 as a phosphate sensor controlling levels of FGF23 is restricted to the bone. A putative role of PiT2 as a phosphate sensor in the intestine, but also in the kidney, requires further investigation. A simple possibility could be that phosphate transport in these organs may be coordinated by PiT2, acting as a sensor, as this is the case in lower organisms [75].

More recently, other phosphate sensors have been identified, adding to the complexity of the overall mechanism of phosphate homeostasis. The calcium-sensing receptor (CaSR) has been found to act as a phosphate sensor in parathyroid cells, controlling PTH synthesis and secretion [76]. This finding originated from the discovery that the CaSR crystal structure has multiple binding sites for phosphate ions in the extracellular domain [77]. Centeno et al. [76] hypothesized that the binding of phosphate could modify the conformation of CaSR and therefore the binding properties of calcium, leading to indirect consequences for the regulation of phosphate homeostasis. They identified that binding of phosphate to arginine residue 62 (R62) inhibited CaSR activity in a non-competitive manner, resulting in increased PTH secretion and explaining the known stimulatory effect of phosphate on PTH secretion, which has an opposite effect to calcium binding. 

The relevance of the CaSR as a phosphate sensor in the kidney is undetermined. The thick ascending limb is the main site of renal CaSR expression [78], where phosphate absorption is negligible. Interestingly, recent studies have demonstrated the expression of NPT2B in this segment, but its function remains unknown [42,43]. In this segment, calcitriol inhibits bicarbonate absorption via a synergic action of aldosterone resulting in stimulation of the ERK pathway [79]. It is possible that a decrease in extracellular phosphate may also act on CaSR to enhance the stimulation of the ERK pathway and to decrease bicarbonate absorption. This may be protective as alkalosis increases phosphate shifts in cells; however, this hypothesis remains to be tested.

The FGF receptor, FGFR1c, was recently shown to be phosphorylated upon increased extracellular phosphate in the absence of canonical ligands in cultured osteoblasts [80]. This phosphorylation led to activation of the MEK/ERK pathway and increased the extracellular levels of FGF23 through an N-acetylgalactosaminyltransferase 3 (Galnt3)-mediated mechanism. Since FGFRs do not bind phosphate [81], it is possible that the “sensing” of extracellular phosphate reported in cultured osteoblasts relates to the ability of phosphate to enter the cell and phosphorylate FGFR1c. If this is the case, FGFR1c may act as a sensor of intracellular rather than extracellular phosphate. It is also possible that FGFR1c and PiT2 functionally interact together to control FGF23 secretion in vivo.

Given that CaSR, FGFR1c, and PiT2 are present in the kidney and small intestine, these proteins may represent a means by which the two organs can sense phosphate levels. While targeting the processes of intestinal and renal phosphate transport to correct phosphate homeostasis disorders is under investigation [82,83], targeting phosphate sensing may also be a novel therapeutic target; although, to date, there are no known PiT2 or FGFR1c modulators. Considering the high homology between PiT1 and PiT2 and the involvement of these proteins in essential processes [32], any target of PiT2 would need to be specific. In the same way, because of the homology and ubiquitous expression of the numerous FGF receptors [84], specific FGFR1c inhibitor may also be hard to develop.

Targeting CaSR in phosphate disorders may also be an interesting approach. In X-linked hypophosphatemia patients, in which secondary hypoparathyroidism can develop because of the necessity for a high phosphate intake [85,86], a single dose of cinacalcet is effective in increasing phosphatemia and TmP/GFR [87]. However, the chronic effect of calcimimetic use in hypophosphatemic disorders is not known. In contrast, a preliminary study from Roberts et al. [88] demonstrated that calcilytics (NPSP795, negative allosteric modulators) can rapidly stimulate PTH secretion but that they do not increase serum calcium in hypoparathyroidism because of CaSR’s gain of function (autosomal dominant hypocalcemia type 1 (ADH1)). The effect of this compound on phosphatemia remains unknown in humans, but it significantly increases phosphatemia in wild-type and ADH1 mouse models [89]. Further studies are needed to assess if calcilytics will find a place in treating phosphatemia related disorders.

## 5. Conclusions

Our understanding of phosphate homeostasis has increased significantly during the last decade, including but not limited to the role of the intestine, FGF23, CaSR, and PiT2 in this process. Understanding organ interactions and the mechanisms involved in phosphate sensing still needs significant research, but it is envisaged that all new knowledge has the potential to be used to develop novel approaches for the treatment of phosphate homeostasis disorders. While it is recognized that improvement in treatment approaches for phosphate imbalance in CKD patients has the potential to significantly enhance their quality of life and reduce mortality, patients with monogenic phosphate disorders and healthy individuals ingesting a highly processed diet are also likely to benefit from a more detailed understanding of the complex processes involved in maintaining phosphate balance.

## Figures and Tables

**Figure 1 ijms-22-05701-f001:**
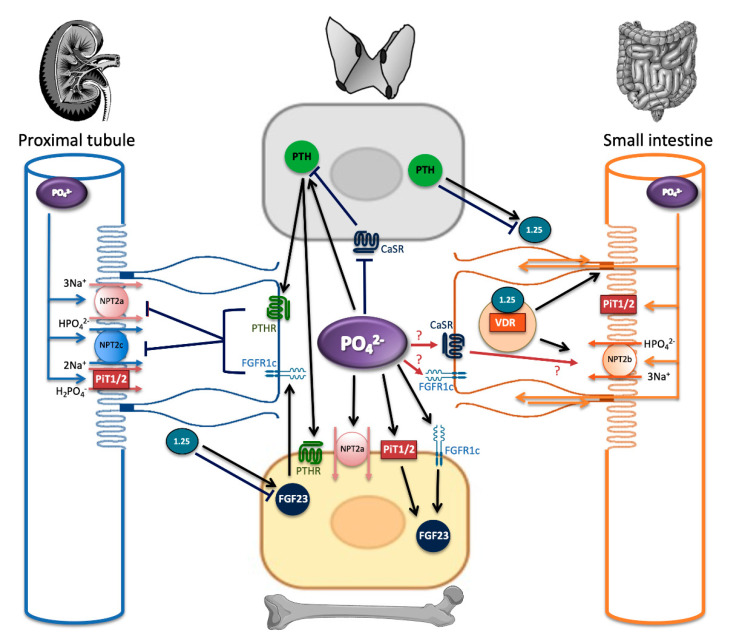
Phosphate homeostasis involves four organs, which interact via the action of the hormones PTH, FGF23, and calcitriol. Osteocytes and osteoblasts secrete FGF23, while proximal tubular cells convert inactive 25(OH)_2_ vitamin D to active calcitriol via the action of the enzyme 1 alpha-hydroxylase [3]. These endocrine hormones regulate the expression and function of tissue-specific phosphate transporters. Extracellular phosphate can also directly act on the putative phosphate sensors, CaSR, PiT2, and FGFR1c to control these processes. Abbreviations: 1.25: 1.25(OH)_2_ vitamin D, PIT1/2: PiT 1 and PiT2, VDR: vitamin D receptor. Red arrows with question marks indicate potential pathways.

**Table 1 ijms-22-05701-t001:** Comparison of human and murine phenotypes following the inactivation of key phosphate transporters and phosphate sensors.

Gene	Human Inactivation	Same Phenotype Human/Murine Y/N	Murine Inactivation
SLC34A1	Idiopathic hypercalcemia [22]	N	Hypophosphatemia High level of calcitriol Retarded secondary ossification [19]
	Hypophosphatemic Nephrolithiasis/osteoporosis [20]	Y	
	Fanconi syndrome [23]	N	
SLC34A2	Normophosphatemia [25]	N	Embryonic lethal E10.5 [26]
		N/A	Conditional knock out: Normophosphatemia Decreased intestinal absorption [27,28]
SLC34A3	Hypophosphatemia Nephrolithiasis Impaired bone function [21,29]	N	Normophosphatemia No bone phenotype [30,31]
SLC20A1	Unknown	N/A	Embryonic lethal E12.5 [32]
SLC20A2	PFBC Normal phosphatemia No data about bone phenotype [33]	Y Y N/A	Brain calcifications Normal phosphatemia Decreased bone quality [34]
CaSR	Severe hyperparathyroidism Osteopenia Failure to thrive [35]	Y	Severe hyperparathyroidism and premature death Rickets [36,37]
	Familial hypocalciuric hypercalcemia [38]	Y	Hypercalcemia Hypocalciuria [36]

Abbreviations: PFBC: primary familial brain calcification, Y: yes, N: no, N/A: not applicable.

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
