# Peer review of "The Complexities of Organ Crosstalk in Phosphate Homeostasis: Time to Put Phosphate Sensing Back in the Limelight"

_ijms, 2021, doi:10.3390/ijms22115701_

Round 1

Reviewer 1 Report

  1. The review is easy to understand but it seems incomplete. A review should cover broad spectrum so that the reader can have an overall idea about the topic. There are few sections that need to be improved.

I would suggest to add sections for

  • Normal phosphate metabolism and how the disruption leads to various diseases.
  • Phosphate metabolism related diseases like chronic kidney disease, cardiac myopathy that makes phosphate metabolism so important
  • Bone mineralization
  • Mechanism how FGF23 plays the important role in phosphate sensing
  • All these topics have recently published good reviews (PMID: 30218014, 26912542, 29580635, 32931449, 33069763, 32101626).
  1. Figure is confusing. I would suggest the authors to use the canonical symbols such as block arrows to show inhibition rather than red pointed arrows because that confuses the reader. There are abbreviations used in the review that are neither used in the review anywhere nor explained in the figure legend such as VDR. that are mentioned in the review.
  2. Table can be improved too by adding more diseases related to the imbalance in the phosphate metabolism.
  3. Or add another table that covers as many human diseases associated with phosphate metabolism as possible so that even if you are unable to cover them in the text, you have reference for every disease.

Reviewer 2 Report

Comments on manuscript ijms-1172360

General Impression:

The review topic is interesting for this field of research. Furthermore, the mechanisms of crosstalk between kidney, parathyroid gland, bone, and small intestine for maintaining phosphate balance are well documented and the corresponding literature is clearly presented and discussed.

Comments and Suggestions for Authors:

The subject of the paper is of scientific interest and deals with the scope of the journal. The manuscript is well written, allowing a quite good easy reading. I recommend a detailed reading since small errors are found in the text.

Minor points:

Minor corrections for improvement are embedded as “sticky notes” in the attached pdf file.
